# Effects of School-Based Exercise and Nutrition Intervention on Body Composition and Physical Fitness in Overweight Adolescent Girls

**DOI:** 10.3390/nu13010238

**Published:** 2021-01-15

**Authors:** Špela Bogataj, Nebojša Trajković, Cristina Cadenas-Sanchez, Vedrana Sember

**Affiliations:** 1Department of Nephrology, University Medical Centre Ljubljana, 1000 Ljubljana, Slovenia; spela.bogataj@kclj.si; 2Faculty of Sport, University of Ljubljana, 1000 Ljubljana, Slovenia; 3Faculty of Sport and Physical Education, University of Niš, 18000 Niš, Serbia; nele_trajce@yahoo.com; 4Institute for Innovation & Sustainable Development in Food Chain (IS-FOOD), Public University of Navarra, 31007 Pamplona, Spain; cristina.cadenas.sanchez@gmail.com

**Keywords:** exercise, diet, adolescents, weight status, fitness

## Abstract

Regular exercise during school hours is encouraged since childhood obesity has reached epidemic proportions. Moreover, a great majority of adolescents do not meet the recommendations for moderate-to-vigorous physical activity. The present study aimed to determine the effects of school-based high-intensity interval training (HIIT) and nutrition intervention on body composition and physical fitness in overweight adolescent girls. Forty-eight girls were included in the study, of whom 24 (age = 15.5 ± 0.7 years) were randomized to a experimental group (EXP) (HIIT and nutrition intervention school-based program) and 24 (age = 15.7 ± 0.6 years) to a control group (CON) that maintained their usual physical education activities. HIIT consisted of 10 stations of own bodyweight exercise and was done three times per week for eight weeks. Moreover, the EXP participated in the nutrition program led by a nutritionist two times a week. Apart from body composition assessment, participants performed countermovement jump (CMJ), medicine ball throw, hand-grip test, and Yo-Yo Intermittent Recovery Level 1 Test (YYIRT1). A significant effect of group (EXP vs. CON) x time (pre vs. post) interaction was observed for weight [F(1,44) = 7.733; *p* = 0.008], body mass index [F(1,44) = 5.764; *p* = 0.020], body fat (in kg) [F(1,44) = 17.850; *p* < 0.001], and body fat (in %) [F(1,44) = 18.821; *p* < 0.001]. Moreover, a significant interaction was observed for the medicine ball throw [F(1,44) = 27.016; *p* < 0.001] and YYIRT1 [F(1,44) = 5.439; *p* = 0.024]. A significant main effect for time was found for hand grip [F(1,44) = 9.300; *p* = 0.004] and CMJ [F(1,44) = 12.866; *p* = 0.001].The present study has demonstrated that just eight weeks school-based HIIT and nutrition intervention, including three sessions a week, can improve body composition and muscular and physical aerobic performance in overweighted adolescent girls.

## 1. Introduction

In the past 20 years, childhood obesity has reached epidemic proportions worldwide [1,2]. In the context of obesity, one of the strategies for tackling this problem is to prolong the time actively spent. However, there is a significant decline in physical activity (PA) during adolescence [3,4,5]. A great majority of adolescents do not meet the recommendations for moderate-to-vigorous PA (MVPA) [6] despite well-documented positive impact on skeletal [7,8], metabolic [9,10], cardiovascular [11], and psychosocial [12] health. In addition, physical fitness (PF) levels appears to be declining in this population [13,14]. Accordingly, childhood and adolescence is a critical period for the development of PF [15] and adolescence has been identified as one of the stages that may play a critical role in the development and persistence of obesity and related comorbidities into adulthood [16]. As the social environment for adolescents diversifies and they become more independent, the key influences on their eating practices begin to change [17]. Therefore, clear strategies to engage adolescents and improve their PA and PF are needed.

Contemporary children often experience a lack of PA and abundance of sedentariness [18] due to a greater amount of time spent in school. Therefore, schools are being recognized as potential effective settings for different exercise or multi-component interventions [19]. A study of 472 youngsters investigated PA levels measured by accelerometry during physical education classes (PE) and showed that subjects spent a reduced amount of time (i.e., less than 50% of the time recommended by PA guidelines) in MVPA, regardless of their weight [20]. A study investigating PA and nutrition behaviors in adolescents concluded that 80% of participants failed to meet guidelines for PA, diet, and sedentary risk behaviors, with girls being less active compared to boys [21]. Systematic review of randomized controlled interventions encompassing nutrition and increased PA in adolescents identified 24 studies evaluating effectiveness of school-based nutrition education in reducing or preventing overweight and obesity [22]. Authors demonstrated reduction of overweight and obesity, as well to increase fruits and vegetable consumption. A more recent review [17] exploring the impact of multi-strategy interventions that encompass nutrition education on adolescents’ health and nutrition concluded significant impact of theoretically-based nutrition education on anthropometric and dietary intake measured in adolescents [17].

In a study with 12 weeks of comprehensive lifestyle intervention [23], 12–16-year-old adolescent boys decreased the body fat (BF) percent; however, these changes did not reflect in the body mass index (BMI). The authors concluded that BMI should not be the only indicator in the assessment of the realization of obesity management intervention. The limitation of the mentioned study was the absence of adolescent girls. In a study with a longer duration (one year) [24], the extracurricular PA program significantly reduced the BMI, adiposity, and improved body composition of adolescents. A 2 week residential summer camp led to improved body weight, BMI, and self-esteem among obese children (9–14 years) [25]. Furthermore, total Impact of Weight on Quality of Life scores improved significantly.

A recent systematic review showed that high-intensity interval training (HIIT) could be a time-effective method for improving health markers in adolescents [26]. The authors found that the effects of HIIT can improve certain aerobic performance parameters in adolescents, whereas improvements in body composition remained unclear. However, some studies showed that adding dietary restriction and behavioral modifications could provoke more benefits for overweight and obese adolescents [27,28,29]. The above-mentioned studies presented interventions that reduced body weight and fat mass while improving PF. Rey et al. (2017) [30] showed that five weeks of vigorous interval training program in combination with a nutrition control improved body composition and PF in obese adolescents. Recently, Plavsic et al. (2020) [31] demonstrated that 12-week HIIT combined with nutrition education led to an improvement in BMI, insulin sensitivity in comparison to the non-exercising group. It was documented that HIIT can improve different aerobic performance parameters in children and adolescents. However, evidence of the positive effects of HIIT combined with nutritional intervention on body composition remains unclear. Another review demonstrated that children have a more significant reduction in BMI z-score in comparison with adolescents, emphasizing the importance of early healthy lifestyle interventions and individual approaches [32]. Despite proven evidence of a benefit of HIIT on a metabolic profile, there is insufficient data in adolescents. Further studies are needed to investigate the effect of HIIT in combination with nutrition interventions on body composition and PF.

The aim of this study was to determine the effects of HIIT and nutrition intervention school-based program on body composition status and PF in overweight adolescent girls. We hypothesized that HIIT, in combination with nutrition intervention, would elicit greater changes in body composition and PF in overweight adolescent girls compared to PE classes only.

## 2. Materials and Methods

This was a randomized, controlled, interventional trial comparing a HIIT and nutrition intervention school-based program with traditional PE in overweight adolescent girls. The flow of the subjects is presented in Figure 1. Forty-eight adolescent girls from different classes in a school from southern Serbia were included in the study, of whom 24 (age = 15.5 ± 0.7 years) were randomized to a HIIT and nutrition intervention school-based program group (EXP) and 24 (age = 15.7 ± 0.6 years) to a control group (CON) that maintained their usual PE activities. There were no time-loss injuries during the intervention. Two participants left the EXP group due to illness (*n* = 1) and missed the final measurement (*n* = 1). Accordingly, a total of 46 participants completed the study, with 22 participants in the EXP group and 24 participants in the CON, which is enough to attain a statistical power of 0.80 (*p* = 0.05).

To be included in the study, participants had to be between 14 and 16 years of age, have a BMI between the 85th and 95th percentile for their gender and age [33], be free of any medications that could affect the results, have no medical problems other than obesity or overweight, and not have participated in any systematic exercise training either at the time of the study or in the previous six months (apart from regular PE at school, which lasts up to 90 min/week).

All participants and parents/guardians were familiar with possible risks associated with the experimental procedures and signed a consent form to participate in the study. The study protocol was conducted in accordance with the Helsinki Declaration and approved by the Faculty of Sports and Physical Education Ethics Committee (Ref. No. 18/2018).

### 2.1. Procedures

The measurement of body composition, muscle mass, and physical aerobic performance of all participants during the initial and final measurements was performed in the school hall. All tests were carried out between the period from 10 a.m. to 1 p.m. at the initial and final measurement and under the same conditions. The same researchers were involved in the initial and final measurements, in the same order and with the same instruments.

All subjects initially performed tests to assess body composition. After which, they performed the PF tests at the stations, with having the physical aerobic performancetest (i.e.,Yo-Yo Intermittent Recovery Level 1 Test, YYIRT1), the last test. The length of the break between tests and within a single test varied according to the complexity and workload of the test.

### 2.2. Physical Fitness Measurements

Body height (in cm to the nearest 0.1 cm) was measured without shoes following standard procedures using a portable stadiometer (Seca 220; Seca Corporation, Hamburg, Germany). Body weight, BMI, BF percentage, BF in kg, and muscle mass were analyzed by a bioelectrical impedance method using a standardized body composition analyzer (Tanita BC 418 MA, Tanita Corp, Tokyo, Japan). Explosive strength of the lower limbs was evaluated with countermovement jump (CMJ) using a sensor mat (PAT 01, Uno lux, Novi Sad, Serbia). The intraclass correlation coefficient (ICC) for test-retest reliability was 0.93. This CMJ was chosen because it is the simplest and the most common in numerous activities. The participant goes from an upright starting position to a half f squat and then jumps up with hands-free to move. In such a jump, it is recommended that the participants leave the ground at the start with their ankles and knees extended and land in a similarly extended position. The task is performed three times with a rest interval of 30 s between jumps, and the best attempt was used for the analysis.

For measuring the strength of upper limbs, hand-grip strength and medicine ball throw were used. Hand-grip strength was measured with the Baseline bulb dynamometer (Baseline, Boise, ID, USA) [34,35]. The reliability of this device was confirmed in our sample with test-retest reliability (ICC = 0.88). The participants were placed in a standard position, feet flat on the floor, sitting upright in a chair with the elbow flexed to 90 degrees as recommended by the American Society of Hand Therapists. When they are ready, the participant squeezes the dynamometer as hard as possible and holds it like this for 2–3 s. Three tests were performed for each hand, and the average value from both hands was used as the result for further analysis. The medicine ball throw was evaluated, with the participant holding a 3 kg rubber medicine ball (Tigar, Pirot, Serbia). The participants were instructed to throw the ball as far as possible straight over their heads. The result is the distance from the front of the line to the point where the ball landed recorded with an accuracy of 1 cm. The best of three attempts was used for further analysis, with a 1-min break between each attempt.

Physical aerobic performance was assessed by YYIRT1, which showed good reliability for 9–16 year old non-athletic children [36]. Briefly, with the sound signal from the audio device, the players run to the cone, which is 20 m away, and back to the start. After arriving at the start, the players have a break of 10 s, during which they have to run slowly to the third cone, which is 5 m away from the start, and back to the start. A new beep follows and players run a distance of 20 m back and forth again (2 × 20 m shuttle run). The running speed increases progressively and is regulated by sound signals that appear at certain intervals. The test was finished when the participants do not complete a shuttle twice within the given signal, and the final score is the total distance covered in meters during YYIRT1.

### 2.3. Exercise Intervention

After the baseline testing, the girls were randomized and assigned to the EXP or the CON in a ratio of 1:1. The EXP received the nutritional intervention and performed a HIIT exercise program three times a week for eight weeks. During this period, the CON engaged only in traditional PE classes. After the intervention, the test procedures were repeated. The endpoint evaluators were blinded to the group allocation. We performed a HIIT, consisting of 10 stations of own body weight exercise, and it was done 3 times per week for eight weeks. Two sessions took place during the PE and one was held outside the PE on different days. We chose a 30 s exercise bout in order to allow at least 15–20 repetitions with appropriate intensity. To achieve maximum time efficiency, a rest period of 15 s was chosen between the exercises. Participants completed two series with a duration of 15 min. Exercises were placed in order that allows resting and working for opposing muscle groups.

Following the warm-up (official PE class), the participants completed the HIIT protocol, which included: (1) push-up on knees, (2) squats, (3) burpees, (4) crunches, (5) step-up, (6) triceps dips, (7) hop jumps, (8) chest medicine ball throws against the wall, (9) skipping rope, and (10) plank. Aiming to avoid boredom and monotony, the order of the exercise was changed after two weeks with respect to the above-mentioned work/rest relationship for muscle groups. Additionally, during the second month, some exercises were exchanged for other exercises that worked the same muscle group (i.e., hop jumps with tuck jumps; crunches with sit-ups).

Two weeks before the training program, participants underwent a familiarization training session in which they learned the technique of exercise as well as the basics of circuit training protocols.

The intensity was controlled with a heart rate monitor (Polar 610i, Finland) during the first and last week of the program in order to ensure the participants were meeting the determined exercise intensity (>80% of their individual maximum heart rate, HR_max_). The HR_max_ was determined during the performance of YYIRT1 [37]. Additionally, the rating of perceived exertion (scale of 0 to 10) was recorded immediately after each session. All workouts were supervised by researchers.

CON was only involved in the traditional PE, which included basketball and volleyball activities during these eight weeks. In addition, the CON also had one extra class for free activities that took place outside the PE.

### 2.4. Nutrition Intervention

Participants from the CON group were advised not to change their eating habits. The EXP group completed the 3-day dietary recall before the start of the intervention. The 3-day dietary recall was used only for information about eating patterns and habits and diet quality to better prepare the diet protocol and to obtain information about possible uncommon habits. The EXP children and their parents participated in the nutrition intervention program led by a nutritionist two times a week. The prescribed diet consisted of 20% proteins, 25–30% fats, and 50–55% carbohydrates. Their calorie intake was between 1300 to 1700 kcal/day based on their body weight. The diet was tailored to the eating habits and target body weight of the individual. The participants were informed about healthy choices, food preparation, regular meals, and portion sizes. They received written instructions and an example of the weekly menu. Each participant was provided with a moderate dietary restriction based on their target body weight (30 kcal/kg x target weight), which was calculated based on their target BMI (50th percentile from the same age category) [33]. The proposed menu was updated weekly throughout the time of the intervention. Adherence to nutrition intervention was controlled once per week by telephone. The distribution of energy intake and meal times were around 30% at breakfast (7:00–7:30), 40% at lunch (12:00–13:00), and 30% at dinner (17:30–18:30). Depending on energy needs, participants were allowed to consume up to two snacks. The most commonly prescribed breakfast choices were cereals with milk and fresh fruit or yogurt with granola and fruits, or two eggs with vegetables (it was mostly replaced with a slice of toasted bread). The most common lunch menu choices consisted of chicken breasts with rice and vegetable, pasta with meat/tomato sauce, turkey with carrots and broccoli, fish fillet with mangold and cooked potatoes, etc. Dinner choices that were most popular among participants were walnut chicken salad, couscous with vegetables, pasta with spinach, carrot pancakes, or sardines on toasted bread. Snacks choices consisted of fresh fruit, cheese and crackers, apple and peanut butter, sliced tomatoes with feta cheese, or a fist of cashews nuts and cranberries. On a daily basis, the girls were recommended to follow the individual menu tailored to their needs, which was designed by the nutritionist following the guidelines from Table 1.

### 2.5. Statistical Analysis

Statistical analysis was performed with the SPSS statistical program version 22 (SPSS Inc., Chicago, IL, United States) and G*Power software version 3.1.9.2 (Die Heinrih-Heine-Universität Düsseldorf, Germany, Europe). The results are presented as mean values ± standard deviation (SD). Kolmogorov–Smirnov test was used to demonstrate that the data had a normal distribution (*p* > 0.05). Furthermore, Levene’s tests were determined for all test variables. A two-way analysis of variance (ANOVA) was used to test the main effect of the group (EXP vs. CON) and the main effect of time (pre-test vs. post-test) and the interaction of group x time for body composition values and PF test results. The magnitude of the Cohen’s d effect (ES) for changes within the group was classified as follows: “trivial” <0.2; “small” 0.2–0.6; “moderate” 0.6–1.2; “large” 1.2–2.0; “very large” >2.0, and “extremely large” >4.0 [38]. A partial eta squared (η^2^) was computed to check the differences between groups where 0.01 was determined as a small effect, 0.06 as a medium effect, and 0.14 as a large effect [39]. Statistical significance was set at *p* ≤ 0.05 level of significance.

## 3. Results

### 3.1. Adherence to Exercise and Nutrition Program

The average adherence to exercise intervention was 92% and 81% for the nutrition intervention. The main reasons for skipping the exercise sessions were menstruation, cold, and fatigue. Non-adherence to the nutrition protocol was mostly related to social events, visiting grandparents, and lack of time for healthy meal preparation.

### 3.2. Body Composition

The body composition results are shown in Table 2. A significant effect of group (EXP vs. CON) x time (pre-vs. post) interaction was revealed for weight [F(1,44) = 7.733; *p* = 0.008], and BMI [F(1,44) = 5.764; *p* = 0.020]. In addition to the significance mentioned above, the significant main effect of time was shown in BF (kg) [F(1,44) = 17.850; *p* < 0.001] and in BF (%) [F(1,44) = 18.821; *p* < 0.001].

### 3.3. Physical Fitness

Table 3 shows the effect of the HIIT and dietary intervention program on PF components. A significant effect of group (EXP vs. CON) x time (pre-vs. post) interaction was revealed for medicine ball throw [F(1,44) = 27.016; *p* < 0.001], and YYIRT1 [F(1,44) = 3.307; *p* = 0.024].

## 4. Discussion

The number of obese children worldwide has risen dramatically in the last twenty years [40], partly due to reduced PA and unhealthy eating habits. A school is a place where adolescents spend most of their day. Therefore, the present study aimed to determine the effects of an 8-week school-based intervention with HIIT and nutrition intervention on body composition and PF in overweight adolescent girls. The main findings of our study were: 1) after eight weeks of HIIT and nutrition intervention school-based intervention, adolescent girls with overweight in EXP reduced their body weight and BMI compared to the CON; and 2) the EXP group improved their upper limbs muscular strength and physical aerobic performance after eight weeks of HIIT and nutrition intervention in comparison with those peers located in CON.

Childhood and adolescent obesity, low physical aerobic performance and decreased PA are associated with various cardiovascular diseases [41]. The results of the current study, although conducted on overweight participants, showed that HIIT and nutrition recommendations were slightly more effective than those of the CON, which is in concordance with earlier statements [42] and recent studies [26,31,42]. However, greater improvements in body composition have been observed in pubertal children [43] compared with prepubertal children, likely due to the maturation effect, duration of interventions (interventions examining pubertal children tend to be longer [31,44] than those in prepubertal children), differences in baseline body mass or body composition, which may confound differences between studies and their attribution to maturity; in addition, most studies of prepubertal children used normal-weight children, in contrast to the studies of pubertal children, which usually used overweight or obese children [26]

Apart from the fact that the current study showed a higher percentage decrease in the percentage of BF and BF in kg in the EXP group compared to the CON, these changes were not sufficient to be statistically significant. A small decrease in BF could be logical considering that study duration has been shown to be a significant moderator of BF, indicating that HIIT interventions longer than eight weeks had greater effects [45] as a shorter intervention duration may lead to erroneous conclusions regarding the efficacy of HIIT in the pubertal population [26]. Recent work [46,47] examined the effects of a HIIT intervention in pubertal children who were divided into a HIIT and a CON group. These authors reported no changes in body composition. Both studies stated that the lack of effect on body composition was due to the short intervention duration (7 and 8 weeks, respectively), while da Silva with colleagues [48] believe that this result was due to the unchanged diet rather than the total duration of the intervention. Further recommendations regarding a minimal HIIT intervention duration remain unclear due to methodological limitations of the studies presented.

Low physical aerobic performance in children and adolescents usually leads to cardiovascular disease later in life [49]. It should be noted that aerobic exercise leads to an increase in aerobic capacity in obese children [42], but greater progress is achieved through the practice of HIIT [50], which can be a key factor of inactivity in children and adolescents, having in mind their lack of time. Our intervention led to an increase in physical aerobic performance. The change in the final test was statistically significant, and the percentage of the group of HIIT was 8.8% (moderate ES). The importance of intensity in improving physical aerobic performance was recently confirmed when obese children who did HIIT made remarkably good progress in aerobic capacity development compared to obese children who exercised free activities for six weeks [51] and after one year long intervention [24]. Similar results were also found in a recent study from Plavsic and colleagues [31]. However, the discrepancy in the results could be due to different methods for determining physical aerobic performance, such as different field tests assessing cardiorespiratory fitness (shuttle run or 1.5 mile run) or gold standard measurements of VO2_max_ (measuring O_2_ and CO_2_ partial pressures in expired air at regular intervals during graded exercise to exhaustion, typically on a treadmill or cycle ergometer) [52].

EXP in our study significantly improved their upper extremity explosive strength, measured by medicine ball throw test. Our results are consistent with the results of a recent study [48], where a twelve-week HIIT program improved the medicine ball throw test result in both normal-weight and overweight obese adolescents. No difference was observed in CMJ after our intervention. Racil et al. (2015) [53] showed that twelve weeks of HIIT and HIIT in combination with plyometric exercises improved CMJ performance in obese young women (16.6 ± 1.3 years, 82.8 ± 5.0 kg). The explanation for this significant change can be found in the fact that their participants had lower initial results (HIIT group: 18.9 ± 2.7 cm and HIIT + plyometric exercises group: 18.8 ± 2.7 cm) compared to our subjects (EXP group: 28.81 ± 4.65). Moreover, the intervention in their study lasted longer (twelve weeks vs. eight weeks). Hand-grip strength was shown to be strongly associated with the cardiovascular health of children and adolescents and underlines the importance of early interventions to target strength preservation and adaptation in primordial prevention [54]. Given the promising results regarding the effectiveness of the school-based HIIT and nutrition intervention program with the traditional PE in overweight adolescent girls, future research should consider incorporating long-term HIIT-based interventions within school settings with a follow-up period to evaluate the long-term positive outcomes of the HIIT and nutrition intervention. Our intervention was not able to improve hand grip strength in our subjects. Therefore, future studies should include longer interventions and exercises targeting upper extremities’ maximum strength.

### 4.1. Practical Implications

Our study has relevant practical implications: firstly, the intervention at the school level is important to increase the PA levels of the children and adolescents who spend most of the time sitting; and secondly, the combined intervention of HIIT and nutrition might be a good strategy to consider for educational policies to improve body composition and PF in children and adolescents.

### 4.2. Study Limitations and Future Research

We recognize possible limitations in the present article: (i) although this study had randomized groups and a controlled period of eight weeks, PA during and outside school was not measured; (ii) only girls were included in the present study, so we cannot generalize the results to both sexes; (iii) the sample size is small, but achieving adequate power statistics of 0.8 (*p* < 0.05); therefore, future studies should be performed on a larger sample size; (iv) intervention was performed on adolescent girls, therefore, results cannot be generalized to samples of varying ages, therefore, future studies should be performed on different ages; (v) furthermore, we did not have a CON that only participated in an exercise program or nutrition intervention; and (vi) dietary intake was assessed with a 3-day dietary recall and the results may be biased, since it is very hard for the participants to remember what did they consume three days ago. That is why we did not include a 3-day recall into the analysis, and it was only meant to assess eating habits and diet quality in order to better plan the nutrition program. Self-reported dietary recalls have been shown to administer under-reported nutrition intake, specifically in obese adolescents [55]; (vii) the use of different field test assessing physical aerobic performance (i.e., shuttle run, 1.5 mile running test) due to better comparisons of cardiovascular health with already published studies, should be use. Nevertheless, this study showed that an eight-week intervention in combination with a short HIIT and nutrition intervention could significantly improve the body composition and PF of overweight girls.

## 5. Conclusions

The present study has demonstrated that just eight weeks of school-based HIIT and nutrition intervention, including three sessions a week, can improve body composition and muscular and physical aerobic performance in overweighted adolescent girls. In particular, this study showed that short-term HIIT implemented in regular PE has the potential to compensate for the deficiencies of the exercise program in a regular class in a very short time. Future research should identify the optimal duration of HIIT and nutrition intervention for improving physical, physiological, and cognitive health in schoolchildren.

## Figures and Tables

**Figure 1 nutrients-13-00238-f001:**
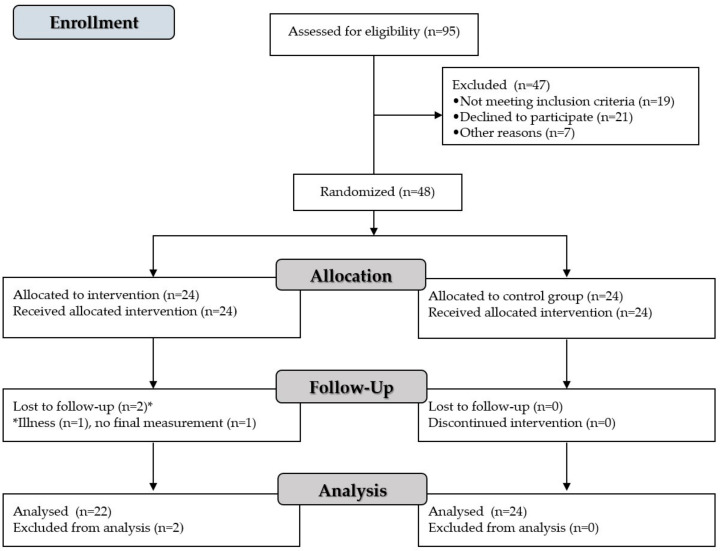
Flow diagram of the study.

**Table 1 nutrients-13-00238-t001:** Proposed portions and items from different food groups.

Food Group	Daily Portions (Units)	Example of Food Items
Cereals and cereal products	9	50 g of rice 60 g of pasta
Vegetables	4	100 g of cabbage, carrots, and leeks
Fruits	3	80 g of bananas 160 g of peaches 300 g of watermelon
Dairy	3	1 unit = 2–2.5 deciliter of milk
Meat or substitutes	6	30 g of lean meat 4 spoons of cooked legumes 3 spoons of cottage cheese 25 g of high-fat cheese 1 egg

**Table 2 nutrients-13-00238-t002:** Effect of exercise and nutrition intervention on body composition and muscle mass components.

Variable	Group	Pre-Test	Post-Test	ES	% Change	*p*-Value, η^2^_p_
Weight (kg)						
	EXP	61.55 ± 4.18	59.20 ± 4.58	−0.54	−3.8%	Group: *p* = 0.596, η^2^_p_: 0.006 Time: *p* < 0.001, η^2^_p_: 0.693 Interaction: *p* = 0.008, η^2^_p_: 0.144
	CON	61.75 ± 4.89	60.41 ± 4.74	−0.28	−2.2%	
BMI (kg/m^2^)						
	EXP	24.42 ± 1.47	23.41 ± 1.41	−0.70	−4.1%	Group: *p* = 0.448, η^2^_p_: 0.013 Time: *p* < 0.001, η^2^_p_: 0.677 Interaction: *p* = 0.020, η^2^_p_: 0.111
	CON	24.59 ± 1.20	23.99 ± 1.66	−0.41	−2.4%	
Muscle mass (kg)						
	EXP	20.42 ± 3.46	20.66 ± 3.60	+0.07	+1.2%	Group: *p* = 0.425, η^2^_p_: 0.014 Time: *p* = 0.393, η^2^_p_: 0.016 Interaction: *p* = 0.393, η^2^_p_: 0.016
	CON	19.70 ± 3.82	19.70 ± 3.70	0.0	0.0%	
BF (kg)						
	EXP	13.73 ± 4.64	12.99 ± 4.39	−0.16	−5.4%	Group: *p* = 0.885, η^2^_p_: 0.000 Time: *p* < 0.001, η^2^_p_: 0.280 Interaction: *p* = 0.080, η^2^_p_: 0.065
	CON	13.33 ± 4.50	13.03 ± 4.14	−0.07	−2.3%	
BF (%)						
	EXP	25.55 ± 5.57	24.84 ± 5.54	−0.13	−2.8%	Group: *p* = 0.414, η^2^_p_: 0.015 Time: *p* < 0.001, η^2^_p_: 0.290 Interaction: *p* = 0.114, η^2^_p_: 0.053
	CON	24.05 ± 5.42	23.73 ± 5.40	−0.13	−1.3%	

Abbreviations: EXP, experimental group; CON, control group; BMI, body mass index; BF (kg), body fat in kg; BF (%), body fat in %; ES, Cohen d effect size.

**Table 3 nutrients-13-00238-t003:** Effect of exercise and nutrition intervention on PF components.

Variable	Group	Pretest	Posttest	ES	% Change	*p*-Value, η^2^*_p_*
HG (psi)						
	EXP	10.82 ± 1.44	11.36 ± 1.47	+0.37	+5.0%	Group: *p* = 0.273, η^2^_p_: 0.026 Time: *p* = 0.004, η^2^_p_: 0.168 Interaction: *p* = 0.949, η^2^*_p_*: 0.000
	CON	11.50 ± 2.69	12.02 ± 2.54	+0.20	+4.5%	
CMJ (cm)						
	EXP	28.81 ± 4.65	30.43 ± 4.48	+0.35	+5.6%	Group: *p* = 0.773, η^2^_p_: 0.002 Time: *p* = 0.001, η^2^_p_: 0.219 Interaction: *p* = 0.182, η^2^*_p_*: 0.038
	CON	28.78 ± 6.23	29.51 ± 7.01	+0.11	+2.5%	
Medicine ball throw (m)						
	EXP	4.61 ± 0.59	5.80 ± 0.60	+2.0	+25.8%	Group: *p* = 0.032, η^2^_p_: 0.096 Time: *p* < 0.001, η^2^_p_: 0.718 Interaction: *p* < 0.001, η^2^_p_: 0.370
	CON	4.65 ± 0.65	5.07 ± 0.52	+0.71	+9.0%	
YYIRT1						
	EXP	1202.73 ± 143.9	1308.18 ± 79.68	+0.91	+8.8%	Group: *p* = 0.438, η^2^_p_: 0.013 Time: *p* < 0.001, η^2^_p_: 0.482 Interaction: *p* = 0.024, η^2^*_p_*: 0.106
	CON	1205.38 ± 132.13	1255.38 ± 104.5	+0.42	+4.1%	

Abbreviations: EXP, experimental group; CON, control group; HG, hand-grip strength; psi, pound per square inch; CMJ, countermovement jump; YYIRT1, Yo-Yo intermittent recovery test level 1; ES, Cohen’s d effect size.

## Data Availability

Data generated and analyzed during this study are included in this article. Additional data are available from the corresponding author on request.

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
