# Peer review of "Effects of School-Based Exercise and Nutrition Intervention on Body Composition and Physical Fitness in Overweight Adolescent Girls"

_nutrients, 2021, doi:10.3390/nu13010238_

Round 1

Reviewer 1 Report

This is an interesting study to identify the impact of school-based exercise and nutrition intervention on body composition and physical fitness in Overweight Adolescent Girls. 

In my opinion, the methodology and results of this study are clear. Discussion is interesting and they used actual references. 

Authors could add some information in the introduction section about sport and nutrition behaviors specifically in adolescents girls.

Author Response

Reviewer 1

This is an interesting study to identify the impact of school-based exercise and nutrition intervention on body composition and physical fitness in Overweight Adolescent Girls. 

In my opinion, the methodology and results of this study are clear. Discussion is interesting and they used actual references. 

Authors could add some information in the introduction section about sport and nutrition behaviors specifically in adolescents girls.

Our response: Thank you for your comment. We have added some information in the introduction section.

Reviewer 2 Report

The manuscript entitled “Effects of school-based exercise and nutrition intervention on body composition and physical fitness in overweight adolescent girls” presents interesting issue, but some areas must be corrected.

Major:

  1. The major limitation of the presented study is associated with very small studied groups (n = 24). Such small studied group does not allow to conclude, as the obtained results are not representative for the general studied group. Taking this into account, the results are seriously biased and Authors are not able to conclude based on them.
  2. Authors declare that they used the method of 3-day dietary recall, which is not a reliable method to assess the dietary intake, as it is very hard for participants to remember what did they consume 3 days ago. A better method is the method of 3-day dietary record (on-going recording for 3 days).

General:

Authors use in their manuscript abbreviations which are not defined while used for a first time or Authors use various definitions (HIIT) – they should correct it carefully in whole manuscript.

Authors should properly formulate their sentences to be clear – e.g. “A significant interaction was revealed for weight” (interaction of what and what?).

Authors should properly use the term of “cardiorespiratory fitness” – in fact they did not analyse it. Authors should get familiar with the possibility to assess cardiorespiratory fitness (e.g. https://www.ahajournals.org/doi/10.1161/CIR.0000000000000866) and properly formulate their observations

Abstract:

Authors should justify their study and present a proper background in this section.

Authors should present specific numeric values that were obtained and the results of the conducted statistical analysis.

Authors should properly formulate conclusions being more general and not formulated only for their program.

Introduction:

Authors should present proper background for their study – as they studied body mass reduction program for children, they should address similar studies of similar programs and present challenges and problems which are defined by other authors (e.g. https://www.ncbi.nlm.nih.gov/pmc/articles/PMC5634063/; https://pubmed.ncbi.nlm.nih.gov/30708984/; https://pubmed.ncbi.nlm.nih.gov/22995865/).

Materials and Methods:

Author should present this section with the necessary details of the program and procedures to be reproducible for other authors.

Authors should provide necessary details of dietary intervention – for the time being there is neither presented how was it planned, nor what it included. Not only general information, but also details are necessary.

Results:

Authors should not reproduce in the text data that are already presented in tables.

Authors should include deepen analysis, to assess the potential influence of interfering factors, etc. as for the time being Authors did not present adequate data to prepare article based on them.

Discussion:

Authors should not reproduce in this section information that are already presented in Introduction and Results Sections.

Authors should present proper discussion for their study – as they studied body mass reduction program for children, they should discuss the obtained results with the results of similar studies of similar programs and discuss challenges and problems which are defined by other authors.

Authors should deepen the discussion of the limitations of their study.

Conclusions:

Authors should properly formulate conclusions being more general and not formulated only for their program.

Round 2

Reviewer 2 Report

The manuscript entitled “Effects of school-based exercise and nutrition intervention on body composition and physical fitness in overweight adolescent girls” presents interesting issue, but some areas must be corrected. Unfortunately Authors did not include corrections based on my previous comments.

Major:

  1. The major limitation of the presented study is associated with very small studied groups (n = 24). Such small studied group does not allow to conclude, as the obtained results are not representative for the general studied group. Taking this into account, the results are seriously biased and Authors are not able to conclude based on them. Authors presented sample size calculation, but only in their response letter, while it should be presented in their Statistical analysis Sub-section.
  2. Authors declare that they used the method of 3-day dietary recall, which is not a reliable method to assess the dietary intake, as it is very hard for participants to remember what did they consume 3 days ago. A better method is the method of 3-day dietary record (on-going recording for 3 days). Based on the answer provided by Authors I have a serious doubts if they are familiar with general nutritional methodology. They answer my comment while referring a position presenting a method of 3-day dietary RECORD, being a totally different method than a 3-day dietary RECALL.

General:

Authors use in their manuscript abbreviations which are not defined while used for a first time or Authors use various definitions (HIIT) – they should correct it carefully in whole manuscript.

Authors should properly use the term of “cardiorespiratory fitness” – in fact they did not analyse it. Authors should get familiar with the possibility to assess cardiorespiratory fitness (e.g. https://www.ahajournals.org/doi/10.1161/CIR.0000000000000866) and properly formulate their observations, as they dod not use the method of the highest availability to assess cardiorespiratory fitness, namely CPET (gas analyzed).

Abstract:

Authors should present specific numeric values that were obtained (and the results of the conducted statistical analysis).

Materials and Methods:

Author should present this section with the necessary details of the program and procedures to be reproducible for other authors. Authors should provide necessary details of dietary intervention – for the time being there is neither presented how was it planned, nor what it included. Not only general information, but also details are necessary (e.g. Authors should justify the system of 3 meals a day)

Results:

Authors should include deepen analysis, to assess the potential influence of interfering factors (e.g. age), etc. as for the time being Authors did not present adequate data to prepare article based on them.

Discussion:

Authors should present proper discussion for their study – as they studied body mass reduction program for children, they should discuss the obtained results with the results of similar studies of similar programs and discuss challenges and problems which are defined by other authors. Authors included some additional sentences, but it is still not enough.
